# Developing ROR1 Targeting CAR-T Cells against Solid Tumors in Preclinical Studies

**DOI:** 10.3390/cancers14153618

**Published:** 2022-07-25

**Authors:** Boon kiat Lee, Yuhua Wan, Zan lynn Chin, Linyan Deng, Mo Deng, Tze ming Leung, Jian Hua, Hua Zhang

**Affiliations:** 1SPH Biotherapeutics (HK) Limited, Unit G01/02, Building 15W, Science Park Phase III, Hong Kong Science and Technology Park, NT, Hong Kong; liwenjie@sphchina.com (B.k.L.); lynnchin@hotmail.co.uk (Z.l.C.); denglinyan@sphchina.com (L.D.); liangzm@sphchina.com (T.m.L.); huaj@sphchina.com (J.H.); 2SPH Biotherapeutics (Shanghai) Limited, Building 9, Cellular Therapeutics Center for Cancers, Gonghexin Road, Shanghai 200072, China; wanyuhua@sphchina.com (Y.W.); dengmo@sphchina.com (M.D.)

**Keywords:** CAR-T-cells, solid tumors, adoptive cell therapy, ROR1

## Abstract

**Simple Summary:**

A ROR1 CAR-T cells derived from scFv of Zilovertamab with short IgG4 hinge spacer region, holds the specific cytotoxicities against a broad range of ROR1^+^ solid tumors with no observed toxicity *in vivo*. Our results have provided preclinical evidence to further develop ROR1 CAR-T adoptive therapy against solid tumors in the clinical stage.

**Abstract:**

Chimeric antigen receptor (CAR)-modified T-cells (CAR-T) have demonstrated promising clinical benefits against B-cell malignancies. Yet, its application for solid tumors is still facing challenges. Unlike haematological cancers, solid tumors often lack good targets, which are ideally expressed on the tumor cells, but not by the normal healthy cells. Fortunately, receptor tyrosine kinase-like orphan receptor 1 (ROR1) is among a few good cancer targets that is aberrantly expressed on various tumors but has a low expression on normal tissue, suggesting it as a good candidate for CAR-T therapy. Here, we constructed two ROR1 CARs with the same antigen recognition domain that was derived from Zilovertamab but differing in hinge regions. Both CARs target ROR1^+^ cancer cells specifically, but CAR with a shorter IgG4 hinge exhibits a higher surface expression and better *in vitro* functionality. We further tested the ROR1 CAR-T in three human solid tumor xenografted mouse models. Our ROR1 CAR-T cells controlled the solid tumor growth without causing any severe toxicity. Our results demonstrated that ROR1 CAR-T derived from Zilovertamab is efficacious and safe to suppress ROR1^+^ solid tumors *in vitro* and *in vivo*, providing a promising therapeutic option for future clinical application.

## 1. Introduction

Chimeric antigen receptors (CARs) are engineered proteins that are expressed on the surface of immune cells, particularly T-cells. CARs are comprised of a tumor antigen recognition domain, a hinge or spacer region, and a transmembrane region, together with the intracellular signaling domain. The antigen-binding domain is usually derived from a single-chain variable fragment (scFv) of an antibody, which enables T-cells to recognize target antigens on tumors in a T-cell receptor (TCR)- and HLA-independent way [1]. The hinge or spacer of CARs links the scFv to the transmembrane domain, providing flexibility to the scFv and improving efficacy [2]. The transmembrane domain connects the extracellular region of CARs to the intracellular signaling domain for CAR activation [2]. The intracellular signaling domain defines the different generation CARs. The first-3ζ signaling domain of the T-cell receptor complex to activate the T-cells. The second- and third-generation CARs include either one or two costimulatory domains, such as CD28 or 4-1BB, OX40, and ICOS which have been shown to promote survival and expansion of the CAR-T-cells in vivo [3,4]. On the basis of the second- and third-generation CARs, the fourth generation “armored” CAR-T further arms T-cells to secrete different molecules, such as cytokines, so-called “truck” or immune-blockers to enhance the T-cell functions [1,4]. 

Despite the promising prolonged clinical response and remarkable effectiveness in hematological malignancies [5], CAR-T therapy still shows limited clinical benefit in solid tumors. One of the major challenges of CAR-T-cell therapy against solid tumors is the lack of tumor-specific antigens that mitigate off-tumor toxicities of immune cells to healthy tissues [6]. Therefore, the ideal tumor targets should be abundantly and uniquely expressed by the neoplasm, with no or little expression in normal cells [6]. 

Receptor tyrosine kinase-like orphan receptor 1 (ROR1) is a Type I surface protein that belongs to the ROR receptor tyrosine kinase family [7]. The human ROR1 protein is composed of an Ig-like (immunoglobulin-like) C2-type domain, a FZ (frizzled) domain, a Kringle domain, and a protein kinase domain [7]. ROR1 is an oncofetal glycoprotein that is expressed during embryogenesis, where it plays a role in differentiation, proliferation, migration, and survival during intrauterine development [8,9]. While ROR1 is usually highly expressed during embryonic and fetal development, it is absent in most adult tissues with some exceptions. For instance, adipose tissue highly expresses ROR1, while the pancreas, gastrointestinal tract, and a subset of immature B-cells have a minor ROR1 expression [10,11]. 

Interestingly, ROR1 is re-expressed in various human tumors. The expression of ROR1 has been reported in numerous blood and solid malignancies and appears to be involved in the inhibition of apoptosis [10,12]. The low or absent expression in normal adult tissues and its high expression in several cancer types have made ROR1 an ideal target candidate for cancer therapy. While some have been working on antibody therapy [13,14,15,16] and antibody-drug conjugates (ADC) [17], there are attempts to utilize ROR1 as a target for CAR-T-cell therapy. Hudecek et al. has reported the first ROR1 CAR-T that utilized anti-ROR1 antibody clone 2A2 and has shown targeted cytotoxicity against primary B-cell chronic leukaemia (CLL) and mantle cell lymphoma (MCL) [11]. The same group further developed another ROR1 CAR-T with scFv that was derived from clone R12, which was then tested in nonhuman primates as they have comparable ROR1 tissue expression to humans and cross-reactivity to human ROR1. Hudecek’s group did not observe any toxicity even at high CAR T-cell doses [18]. However, when testing another CAR-T with scFv that was derived from clone R11 antibody, which targets the Kringle domain of ROR1 that is conserved in humans and mice, they observed lethal toxicity in mice [19]. These findings highlighted the importance of the antigen recognition domain in the development of a safer ROR1 CAR-T. 

In this study, we employed the scFv that was derived from Zilovertamab (formerly called Cirmtuzumab or UC-961) as the antigen recognition domain of CARs. Zilovertamab is a clinical stage humanized monoclonal antibody. Preclinical studies showed that Zilovertamab did not cross-react with human post-partum tissues, including pancreatic and adipose tissue that have been reported with ROR1 expression [20]. Zilovertamab is currently under Phase ½ clinical trial, preliminary results showed that it is well-tolerated and responsive in patients with MCL or ALL in combination with the BTK inhibitor ibrutinib [21]. Here, we aim to assess the functionality of two Zilovertamab-derived ROR1 CARs, which have different spacer domains, against various tumors. We first tested the ROR1 CAR-T-cell activation and cytotoxicity in vitro before further evaluating the in vivo functionality of ROR1 CAR-T in three solid tumor xenografted mouse models.

## 2. Materials and Methods

### 2.1. Cell Lines

HEK293T, Jurkat-E6, Jeko-1 were purchased from American Type Culture Collection (ATCC, Manassas, VA, USA). Cancer cell lines A549, MDA-MB-231, NCI-H1975, K-562 and Nalm6 were kindly provided by Professor Xin-yuan Guan (HKU). Cancer cell lines MEC-1 and MEC-ROR-1 were gifts from Prof. Charles E. Prussak of UCSD. The Jurkat-lucia cell line was purchased from Invivogen. The GFP-expressing cell lines were generated with lentivirus, pALD-LentiEGFP vectors (Aldevron, Fargo, ND, USA), that carried GFP genes and sorted in a 96-well plate for single cell cloning. These cell lines were cultured in complete IMDM or DMEM media (10% heat-inactivated FBS, and 1% penicillin-streptomycin (P/S)) (Invitrogen, Waltham, MA, USA).

### 2.2. CAR Constructs and Lentivirus Preparation

The genes of ROR-1 CARs comprising of a single chain variable region (scFv) from Zilovertamab, a short IgG4 Fc hinge or a long IgG4 Fc hinge-CH2-CH3 spacer, CD28 transmembrane domain, and an intracellular signaling domain contains 4-1BB-CD3zeta were synthesized by Genewiz (Suzhou, Jiangsu, China). The genes were cloned into pALD-LentiEGFP vectors (Aldevron) by recombinant cloning via Apa1/Nhe1 cloning sites using Hyper Assembly Cloning Kit (APEXBIO Tech, Houston, TX, USA) following the protocol. The GFP marker was removed from this cloning process. Lentivirus were prepared in HEK293T-cells by transduction with a four-plasmid system coding for the lentivector genomes, pALD-VSV-G, pALD-Rev, and pALD-GagPol (Aldevron) using Lipofectamin 3000 (Invitrogen). The lentivirus was harvested at 48 and 72 h after transduction and concentrated by an Amicon Ultra-15 100 kDa MWCO centrifugal concentrator (MilliporeSigma, MA, USA) if necessary. All the virus preparations were frozen at −80 °C for further experimentation.

### 2.3. Generation of ROR-1 CAR-T

Human peripheral blood mononuclear cells (PBMCs) were stimulated with a T-cell activation/expansion kit (Miltenyi Biotech) at a 1:1 (beads to cells) ratio for 2–3 days and grown in the presence of IL-2 (500 IU/mL) in complete AIM-V media (10% FBS, 1% P/S) before being transduced with lentivirus. T-cells or Jurkat cells were transduced with polybrene (8 ug/mL) (Santa Cruz, CA, USA) and concentrated lentivirus using spinoculation. The transduction efficiency was determined on days 2–5 post-transduction by flow cytometry using an FITC-conjugated recombinant human ROR1 protein. The T-cells or Jurkat cells were further expanded for 2 weeks or less.

### 2.4. LDH Assay

The LDH assay was carried out with CyQUANT^TM^ LDH Cytotoxicity Assay (Invitrogen) following the protocol. In brief, the T-cells were seeded at 2 × 10^4^ per well, while the target cells, MEC-1 and MEC-ROR1 cells were then added according to the effector to target ratio of 1:1 or 1:3. An equal number of the target cells were also plated in the different wells to measure either the spontaneous LDH activity or the maximum LDH activity with cell lysis buffer. The cytotoxicity was calculated with the following formula:$$\mathrm{Cytotoxicity}\;\left(\%\right)=\left[\frac{Cell\;induced\;LDH\;activity-Sponteneous\;LDH\;activity}{Maximum\;LDH\;activity-Spontaneous\;LDH\;activity}\right]\times 100\%$$

### 2.5. Flow Cytometry

The surface expression of ROR1 CAR was determined using conjugated protein FITC-ROR1-his tag (Acrobiosystems, Newark, DE, USA). The following conjugated antibodies were used to identify the T-cell populations: anti-CD3, anti-CD4, and anti-CD8 (Biolegend, San Diego, CA, USA). Anti-ROR1 (clone 4A5, BD Biosciences, Dubai, UAE) was used to identify the ROR1 expression on tumor cell lines. Anti-CD137 (Biolegend) was used to evaluate the upregulation of CD137 on CAR-T-cells. All the antibodies or proteins were used at the suggested concentration. The staining procedure began with collecting the cells and washing once with FACS buffer (0.1% sodium azide + 2 mM EDTA + 0.5%BSA) before adding the conjugated antibody. The cells were incubated with antibodies at 4 °C for at least 30 min, and then 7-AAD was added and incubated for 5 min at 4 °C. The cells were washed once with FACS buffer and were ready for flow cytometer analysis. All the staining was performed with fluorochromes-conjugated antibodies/proteins without secondary antibodies staining. No cell fixation was involved for all staining. We collected at least 10,000 cells from the gated population in FSC versus SSC plot for analysis. The data were acquired on CytoFlex LX (Beckman Coulter, Brea, CA, USA) and analyzed using FlowJo v.10.7 (FlowJo).

### 2.6. Animal Protocol

The animal studies were performed by PharmaLegacy Laboratories (Shanghai, China). NOD.scid.γc-/- (NSG) mice were purchased from Beijing Vitalstar Biotechnology under PharmaLegacy IACUC-approved protocols. The mice were kept in a specific-pathogen-free (SPF) environment in individual ventilated cages with 4–6 mice per cage, bred with sterilized food and water, and maintained on a 12 h light–12 h dark cycle.

In brief, 1 × 10^6^ Jeko-1-luc cells were delivered intravenously (i.v.) whereas the MD-MBA-231 or NCI-H1975 tumor cells were mixed with Matrigel and injected subcutaneously (s.c.) into the mice. A total of three days after the tumor implantation, a group of 5–6 mice received a single dose of 1 × 10^6^, 2 × 10^6^ or 3 × 10^6^ CAR-T or Mock-T-cells. The mice received 6 × 10^4^ IU IL-2 intraperitoneal (i.p.) every 2 days. The weight of the mice, tumor size, or bioluminescence were measured at designated dates. 

### 2.7. Statistical Analyses

Prism 9 Software (GraphPad, San Diego, CA, USA) was used to conduct statistical analyses. A Student’s *t* test was performed as a two-tailed paired test with a confidence interval of 95% and results with a *p*-value that was less than 0.05 were considered significant. Statistical analyses of survival were conducted by log-rank testing and the results with a *p*-value that was less than 0.05 were considered significant.

## 3. Results

### 3.1. ROR1 Is Expressed on Various Human Tumor Cell Lines

We investigated the expression of ROR1 antigen on five human cancer cell lines, including human lung cancer cell A549, HCC827 and NCI-H1975, human breast cancer cell line MDA-MB-231, and human mantle cell leukaemia (MCL) cell line Jeko-1. The B-chronic lymphocytic leukaemia (B-CLL) MEC-1 cell line and the B-acute lymphocytic leukaemia (B-ALL) Nalm 6 cell line that do not express ROR1 served as negative controls while MEC-1 that was genetically transduced to express the ROR1 gene (human ROR NM_005012.4), namely MEC-ROR1, was a positive control (Figure 1). All human cancer cell lines that were tested expressed ROR1 antigen at different levels and thus could be a target of ROR1 CAR-T in the study.

### 3.2. ROR1 CAR with Shorter Spacer Region Exhibited Superior Expression

We engineered two ROR1 CAR constructs. Both are second-generation CARs with a 4-1BB-CD3z signaling domain but contain a major difference in their spacer region. The Hinge CAR contained a short IgG4 hinge region while the CH3 CAR was with a long IgG4-CH2-CH3 hinge region (Figure 2A). We first tested the expression of CARs on the Jurkat-lucia cell line (Invivogen, San Diego, CA, USA), a human T-lymphocyte-based Jurkat cell line that is integrated with an NFAT-inducible Lucia reporter gene. We noticed that the CH3 ROR1 CAR had consistently exhibited a lower expression rate compared to the Hinge ROR1 CAR (Figure 2B,C). In a consensus, the optimal spacer length of a given CAR depends on the position of the targeted epitope [22]. Therefore, a longer spacer only provides extra flexibility to the CAR and allows for better access to membrane-proximal epitopes. Given the fact of the higher ROR1 Hinge CAR expression here and the fact that Zilovertamab has the specificity for a distinctive epitope in the distal extracellular domain of human ROR1 [20,23], we thereby selected the ROR1 Hinge CAR construct for further studies.

### 3.3. ROR1 Hinge CAR-T-Cells Demonstrated Potent Anti-Tumor Activities

To further investigate into the in vitro functionality, we transduced the ROR1 Hinge CAR into primary human PBMC for expression (Figure 3A). The expression level of ROR1 Hinge CAR on PBMC was similar to those on Jurkat-lucia. We tested the cytotoxicity of ROR1 Hinge CAR-T against the ROR1^+^ MCL cell line Jeko-1 that was engineered to express GFP. The ROR1 Hinge CAR-T-cells eliminated the Jeko-1 population after a 48-h of the co-culture as measured by flow cytometry, suggesting its specific cytotoxicity against ROR1 expressing target cells (Figure 3B). We further tested a list of cancer cell lines that express the ROR1 antigen and with a GFP reporter marker. The ROR1 Hinge CAR-T-cells exhibited cytotoxicity against breast cancer cell line MDA-MB-231, lung cancer cell line HCC827, and NCI-H1975, demonstrating the ROR1 Hinge CAR-T was effective against different types of tumors (Figure 3C). Furthermore, the cytotoxicity of ROR1 Hinge CAR was also confirmed with a Lactate Dehydrogenase (LDH) activity assay. The ROR1 Hinge CAR-T-cells induced high cytotoxicity against the ROR1^+^ MEC-ROR1 cell line while showing no effect on the ROR1-MEC-1 cell line (Figure 3D). We assessed the 4-1BB expression on ROR1 Hinge CAR-T after incubating with different tumor cells. 4-1BB, also known as CD137, is a member of the TNF receptor superfamily with T-cell costimulatory functions. Naive T-cells do not express 4-1BB. However, upon stimulation, activated T-cells transiently express a high level of 4-1BB, which disappears rapidly. The ROR1 Hinge CAR-T-cells themselves did not induce 4-1BB expression and the co-culture with MEC-1, which is a ROR1-cell line, did not upregulate the 4-1BB. However, after stimulation of the ROR1^+^ target cells such as A549, HCC827, MDA-MB231, and MEC-ROR1, 4-1BB was upregulated on both CD4^+^ and CD8^+^ ROR1 Hinge CAR-T-cells (Figure 3E). In short, our data suggest that ROR1 Hinge CAR could target and kill various ROR1-expressing tumor cells. 

### 3.4. ROR1 Hinge CAR Cells also Demonstrated Promising Effect on Murine Solid Tumor Models

We then tested the ROR1 Hinge CAR cells in vivo. We first evaluated the ROR1 Hinge CAR-T-cells against MCL. NOD.scid.γc-/- (NSG) mice were intravenously injected with luciferase-positive Jeko-1 tumor cells. A total of seven days later, the tumor burden was evaluated based on bioluminescent imaging (BLI), and a group of five mice received a single dose of 1 × 10^6^ or 2 × 10^6^ of either Mock-T or ROR1 Hinge CAR-T through tail vein injection. A total of seven days after the T-cell injection, the mice that received either a high or low dose of ROR1 Hinge CAR-T-cells controlled the tumor growth significantly compared to the untreated mice. The significant tumor suppression effect against the untreated mice was persistent up till 28 days in both groups of mice that received ROR1 Hinge CAR-T treatment (Figure 4A top panel). 

We further evaluated the anti-tumor effect of ROR1 Hinge CAR-T against other solid tumor models. To assess the efficacy of ROR1 Hinge CAR-T, two additional solid tumor xenograft models in NSG mice were established. The NSG mice were engrafted with 1 × 10^6^ of MDA-MB-231 breast cancer cells or NCI-H1975 lung cancer cells intraperitoneally. A single dose of ROR1 Hinge CAR-T-cells or Mock-T-cells with different cell numbers was given on day 7 after the engraftment. The tumor volume was measured in 2–3 day intervals after the treatment. 

In the breast cancer MDA-MD-231 mouse model, the mice receiving a high dose (3 × 10^6^) of the CAR-T-cells significantly suppressed tumor growth compared to the untreated mice (Figure 4B top panel). The tumor remained significantly suppressed for up to 37 days. When the mice were sacrificed on day 38 post-tumor engraftment, those mice that received ROR1 Hinge CAR-T-cell treatment had significant lower tumor weight compared to the untreated mice (Figure 4D). 

Meanwhile, in the lung cancer NCI-H1975 model, the mice received 2 × 10^6^ cells of Mock-T or ROR1 Hinge CAR-T-cells. The mice that received ROR1 Hinge CAR-T exhibited superiority in controlling tumor growth compared to the mice that received the mock T-cells or the untreated mice (Figure 4C top panel). Consistently, when the mice were sacrificed 29 days after the engraftment of tumors, the ROR1 Hinge CAR-T-treated mice had a smaller tumor compared to the untreated group (Figure 4E). Our data support the hypothesis that ROR1 Hinge CAR-T-cells can efficiently suppress the growth of different solid tumors in mice. 

In addition to the above in vivo model studies, we did not observe obvious toxicity that was induced by the CAR-T treatment, such as dramatic weight loss (Figure 4A–C bottom panels). Neither did we observe the differences of gross appearance or behavior between the treated and control mice. These results indicate that ROR1 Hinge CAR could suppress the tumor growth without obvious side effects.

## 4. Discussion

While CAR-T treatment has shown striking achievement against hematological malignancies [5], there are challenges that remain to be overcome in CAR-T-cell therapy against solid tumors [6]. One of the main dissimilarities between blood cancers and solid tumors is the expression of tumor antigen. Blood cancer commonly expresses unique and individual markers, but solid tumors usually do not contain a specific tumor marker. It is more common to recognize a tumor-associated antigen (TAA) on solid tumors, such as GD2, mesothelin, and PSMA etc., where the expression is increased on cancerous cells. However, those TAAs are also expressed on normal tissues to a much lower degree and thus might cause severe on-target off-tumor toxicity. Hence, those aberrantly or overexpressed antigens on solid tumors that are also expressed on normal tissues must be carefully evaluated to be a target for CAR-T-cell therapy [6]. In this study, we constructed a CAR targeting ROR1, which has been reported to be expressed at very low levels on some human tissue but aberrantly expressed on various tumor cells [10]. Our ROR1 CAR utilized the scFv of Zilovertamab, a clinical-stage monoclonal antibody that has been shown to be safe and effective against MCL and ALL in a Phase ½ clinical trials [21].

We compared two different CAR constructs with different lengths of spacer region. The spacer region of a CAR has been proven to affect the CAR expression, recognition, and effectiveness [2]. Previously, Hudecek et al. have demonstrated that ROR1 CAR-T with scFv that was derived either from 2A2 or R12 antibodies, which recognized the Ig-like/Frizzled region of ROR1, conferred optimal T-cell recognition and function with the short IgG4 Hinge linker [24]. When compared with the short IgG4 Hinge and the long IgG4-CH2-CH3 spacer on our ROR1 CAR-T, we observed similar results. The short IgG4 Hinge exhibited higher expression and better functionality compared to the long IgG4-CH2-CH3 hinge.

We showed that ROR1 Hinge CAR was expressed on both Jurkat-lucia and human primary T-cells. Importantly, the ROR1 Hinge CAR exhibited antigen specificity against the ROR1^+^ cell lines and activated upon the stimulation of ROR1^+^ cell lines but not against ROR1-cell lines. We tested ROR1 Hinge CAR against lymphoma in vivo. The CAR was able to significantly suppress the tumor growth in Jeko-1 cell-engrafted mice. While the efficacy of CAR-T in solid tumor remained a challenge, our ROR1 Hinge CAR-T-cells significantly suppressed tumor growth compared to the control groups in different solid tumor xenograft models, suggesting the ROR1 Hinge CAR is also effective against different types of solid tumors. Importantly, we did not observe an obvious weight loss in the mice that received a high dose of ROR1 CAR-T-cells, indicating the ROR1 Hinge CAR is safe and well-tolerated. Previously, Hudecek et al. reported that two different ROR1 CARs controlled the CLL and MCL cancer growth in vivo, while Wallstabe et al. demonstrated the effectiveness of ROR1 CAR-T-cells on solid tumors using a 3D cancer model [24,25]. Despite having different antigen recognition domains, our results are consistent with their findings. Furthermore, our in vivo data provided direct efficacy evidence of ROR1 CAR-T against various solid tumors. Although ROR1 is still being evaluated as a CAR target in a clinical trial (NCT02706392) in lung and breast cancer, the clinical efficacy has been very limited [26]. This might be due to the differences between the epitopes that they targeted [27,28] or the higher affinity of different scFv leading to T-cells themselves exhausted and died [29]. 

In summary, we here have developed a second generation ROR1 CAR-T using 4-1BB signaling domain, derived from scFv of Zilovertamab with a short IgG4 hinge spacer region, in response to the lack of the progress of CAR T-cells in the treatment of solid tumors. It demonstrates the specificity against a broad range of ROR1^+^ solid tumors with no observed toxicity in vivo. Our results have provided preclinical rationale to further develop ROR1 CAR-T adoptive therapy against solid tumors in the clinical stage. 

## 5. Conclusions

Here, we constructed and compared two CAR targeting ROR1 with different lengths of spacer region. However, we observed the short IgG4 Hinge exhibited higher expression and better functionality in vitro compared to the long IgG4-CH2-CH3 Hinge CAR. Further, in different solid tumor xenograft models, our ROR1 Hinge CAR-T cells significantly suppressed tumor growth compared to the control groups without seen toxicity, suggesting the ROR1 Hinge CAR is safe and effective against different types of solid tumors.

## Figures and Tables

**Figure 1 cancers-14-03618-f001:**
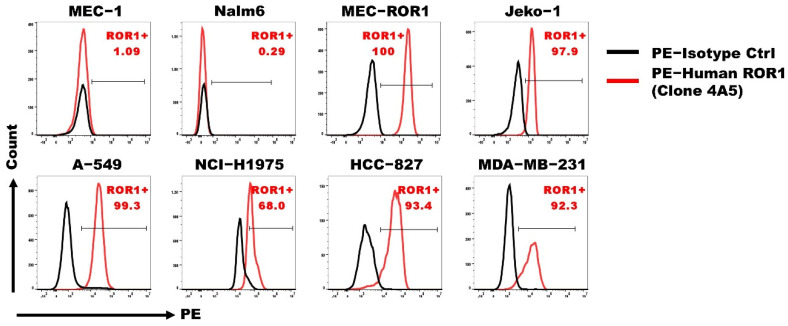
Various human cancer cell lines that expressed the ROR1 antigen. Lung cancer cell line A549, HCC827, H1975, breast cancer cell line MB-MDA-231, and mantle cell lymphoma cell line Jeko-1 were stained with anti-ROR1 antibody (clone 4A5) or isotype control and analyzed by flow cytometry. B-chronic lymphocytic leukemia cell line MEC-1 and B-acute lymphocytic leukemia cell line Nalm 6 served as negative controls, whereas MEC-1 that was transduced to express the ROR1 antigen (MEC-ROR1) served as a positive control. The black solid line represents an isotype control, while the red solid line represents the antibody staining.

**Figure 2 cancers-14-03618-f002:**
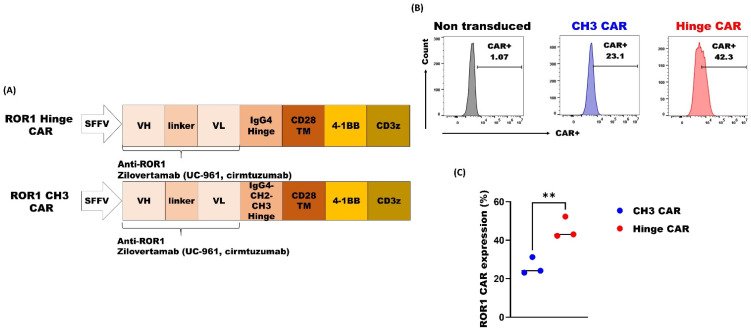
ROR1 CAR with shorter spacer exhibited superior expression. (**A**) The schematic ROR1 Hinge CAR and ROR1 CH3 CAR constructs; (**B**) The expression of CARs was detected with recombinant ROR1 protein on Jurkat-lucia 3 days after lentivirus transduction and analyzed by flow cytometry, a representative of more than 3 experiments; (**C**) ROR1 CAR expression rates from 3 independent experiments (** *p* < 0.01).

**Figure 3 cancers-14-03618-f003:**
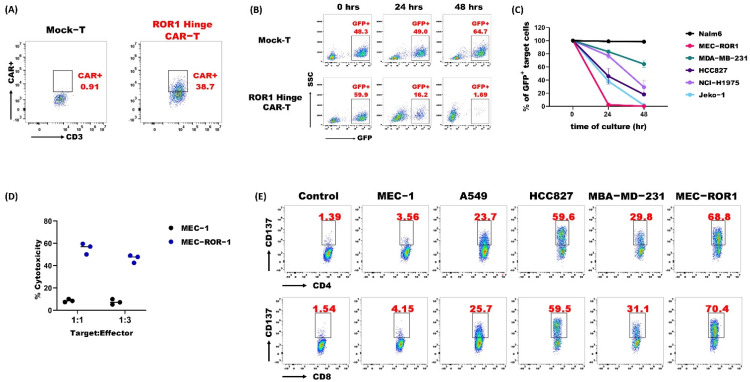
ROR1 Hinge CAR-T-cells show potent in vitro activities against various tumor cell lines. (**A**) The expression of ROR1 Hinge CAR on T-cells that were derived from PBMC at day 5 after lentivirus transduction was detected by flow cytometry; (**B**) Jeko-1 with GFP marker was co-cultured with Mock-T or ROR1 Hinge CAR-T at 1:1 ratio. The population of GFP-expressing cells at 0 h, 24 h, and 48 h were detected by flow cytometry; (**C**) Various cancer cells that express GFP were co-cultured with ROR1 Hinge CAR-T-cells at 1:1 ratio. The GFP-expressing cells were detected by flow cytometry at 0 h, 24 h, and 48 h; (**D**) ROR1 Hinge CAR-T-cells co-cultured with MEC-1 or MEC-ROR1 and cellular cytotoxicity of CAR-T was determined by LDH cytotoxicity assay at different E:T ratios; (**E**) ROR1 Hinge CAR-T-cells were co-cultured with different tumor cells at 1:1 ratio for 24 h, then CD137 expression level was detected by flow cytometry. All the figures shown here are a representative of more than 3 experiments.

**Figure 4 cancers-14-03618-f004:**
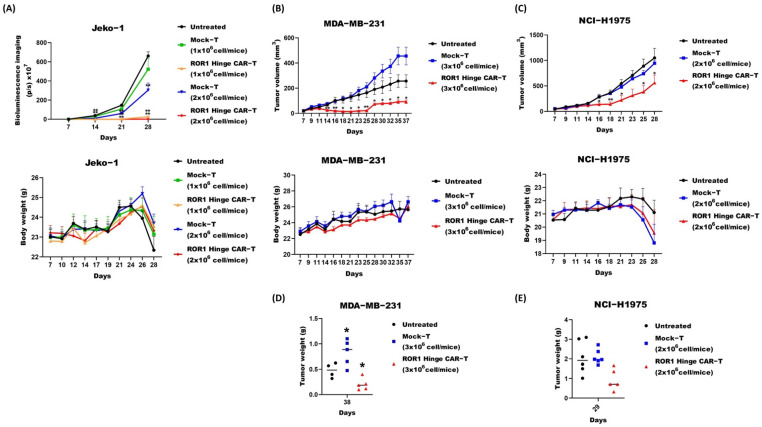
ROR1 Hinge CAR-T-cells suppressed solid tumor growth and posed no obvious side effects in vivo. (**A**) Jeko-1 animal model, the NSG mice that received 1 × 10^6^ Jeko-1 that was transduced with luciferase construct intravenously. At day 7, the mice received a single dose of different numbers of Mock-T or ROR1 Hinge CAR-T-cells. Bioluminescence (**Top**) was measured once a week, whereas body weight (**Bottom**) was measured at designated dates (n = 5 or 6, * *p* < 0.05, ** *p* < 0.01); (**B**) MDA-MB231 animal model, the NSG mice received 1 × 10^6^ MDA-MB-231 cells subcutaneously at one side. At day 3, the mice were treated with a single dose of Mock-T or ROR1 Hinge CAR-T-cells at the designated numbers. Tumor volumes (**Top**) were measured over time (n = 5 or 6, * *p* < 0.05, ** *p* < 0.01), whereas body weight (**Bottom**) was measured at designated dates; (**C**) NIH-H1975 animal model, the NSG mice received 1 × 10^6^ NIH-H1975 cells subcutaneously at one side. At day 3, the mice were treated with a single dose of Mock-T or ROR1 Hinge CAR-T-cells at the designated numbers. The tumor volumes (**Top**) were measured over time (n = 5 or 6, * *p* < 0.05, ** *p* < 0.01), whereas body weight (**Bottom**) was measured at designated dates; (**D**) Mice bearing MDA-MB-231 tumor were sacrificed at day 38, the tumors were extracted and weighed (n = 5 or 6, * *p* < 0.05, ** *p* < 0.01); (**E**) The mice bearing NIH-H1975 tumor were sacrificed at day 29 when the tumors were extracted and weighed (n = 5 or 6, * *p* < 0.05, ** *p* < 0.01).

## Data Availability

The data used and/or analyzed during this study are available from the corresponding author upon request.

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
