# Peer review of "Developing ROR1 Targeting CAR-T Cells against Solid Tumors in Preclinical Studies"

_cancers, 2022, doi:10.3390/cancers14153618_

Round 1

Reviewer 1 Report

The manuscript by Lee and co-workers investigates the therapeutic action of ROR1 targeting CAR-T cells for solid tumor in a preclinical setting. The study aims at extending CAR-T applications to solid tumors and studies ROR1 CAR-T cells for specificity and in vitro cytotoxicity. The in vivo tumor control potential of the CAR-T cells is tested in three solid tumor xenograft mouse models.

Comments:

The manuscript shows rather remarkable in vivo suppression of solid tumor growth by the CAR-T cells in the NSG mouse models. However, how this relates to the characteristics of the CAR-T cells is less obvious. How were the CAR-T cells expanded following the lentiviral transduction ie which T-cell subsets expressing the CAR construct are used for the in vitro and in vivo experiments? Were the CAR-T cells enriched for a specific cell population or were T-cells expanded as bulk. How does the CAR expression following expansion relate to the expression at day 3 or 5 after transduction (as shown in figures 2 and 3)? Currently there exists an apparent discrepancy between the figure 3A and 3E results with higher proportion of T-cells becoming active than expressing the CAR construct in relation to the target stimulus.

The manuscript would also benefit if the properties of the CAR-T cells related to their cytotoxic activity were elaborated beyond the current assessment of CD137/4-1BB expression. Do the ROR1 Hinge CAR-T cells show altered proliferative capacity or cytokine and eg. granzyme production following stimulation by the target cells? How does this compare to the initial level of antigen stimulus or level of CAR expression in a specific T-cell population? Is this recapitulated in the in vivo setting?

General comments on figures:

The resolution in many figures could be better to allow viewing of enlarged images. This is especially important for figure 3 where flow cytometry data is shown as dot plots.

Author Response

Reviewer #1 (Reviewer Comments to the Author):

The manuscript shows rather remarkable in vivo suppression of solid tumor growth by the CAR-T cells in the NSG mouse models.

We are very encouraged by the positive comments and are grateful for the critical evaluation for improvement.

> However, how this relates to the characteristics of the CAR-T cells is less obvious.

Our study showed the 2nd generation ROR1 CAR-T cells with a shorter hinge region can treat and inhibit the ROR1+ tumor growth in the xenografted tumor models.

> How were the CAR-T cells expanded following the lentiviral transduction ie which T-cell subsets expressing the CAR construct are used for the in vitro and in vivo experiments? Were the CAR-T cells enriched for a specific cell population or were T-cells expanded as bulk.

Here we transfected and expanded the bulk PBMC with the lenti virus to conduct in vitro and in vivo experiments. Both CD4 and CD8 were expressing ROR1 CAR. However, we didn’t purposely enrich for specific T cell subsets for this study.

>How does the CAR expression following expansion relate to the expression at day 3 or 5 after transduction (as shown in figures 2 and 3)?

We know the T cells, after activation and transfection, would experience a brief contraction and long expansion growth curve, lasting up to 10-14days. Therefore, day 3 and 5 expression rates of the CAR may have some differences. However, we have found that the expressions at day 3 or 5 doesn’t show much big difference as other non-transfected cells also expand as well.

>Currently there exists an apparent discrepancy between the figure 3A and 3E results with higher proportion of T-cells becoming active than expressing the CAR construct in relation to the target stimulus.

4-1BB expression is commonly used as a T cell activation marker and a functional marker like CD107a. Figure 3A showed the plot of CD3 expressing ROR1+ CAR, whereas Figure 3E showed 4-1BB expression by CD4+ and CD8+ cells respectively after co-cultures with the tumor targets. So, there should be no discrepancy between CAR and 4-1BB expression according to the different gating strategies and the cell population plots used in those figures.

>The manuscript would also benefit if the properties of the CAR-T cells related to their cytotoxic activity were elaborated beyond the current assessment of CD137/4-1BB expression.

A great point! We could do more various assays for analysis, such as 51Cr release assay, interferon ϒ release assay, real-time quantitative live-cell imaging, etc. Due to limited resources in our lab, we just used the equipment we could access, such as flow machine, ELISA plate reader, luminometer, etc.

>Do the ROR1 Hinge CAR-T cells show altered proliferative capacity or cytokine and eg. granzyme production following stimulation by the target cells?

These are good points to delineate their biological differences! However, we didn’t compare the hinge with the ch3 versions in those biological properties. We merely investigated whether it had a better surface expression of the CAR and considered the fact of the antibody targeting the distal region of the extracellular domain of ROR1, thus determining which one would be better to go further to the clinical. Moreover, since the CH3 version contained a larger transgene segment, we guessed that made it less efficient in virus packaging, resulting in the lower CAR expression.

>How does this compare to the initial level of antigen stimulus or level of CAR expression in a specific T-cell population? Is this recapitulated in the in vivo setting?

We followed the same protocol to stimulate bulk PBMC with anti-CD3/CD28 beads in both in vitro and in vivo studies. We usually got 20-60% CAR expression in both of those settings.

> General comments on figures: The resolution in many figures could be better to allow viewing of enlarged images. This is especially important for figure 3 where flow cytometry data is shown as dot plots.

Sure, thank you. We have made them better.

Reviewer 2 Report

1. The authors should provide a Simple Summary of the research study articulating the rationale of the research question and the findings.

2. IL-2 ELISA assay: There are no data presented in the manuscript.

3. Luciferase reporter assay: There are no data presented in the manuscript.

4. M&M Flow cytometry: What is the final cell counted? Are all subsequent steps were done on ice? How many times the cells were washed before primary antibody addition? What concentration of primary Abs was used to treat the cells and incubated for how long? The secondary control cells were treated with the same? What secondary Ab at a final concentration was used? Where the cells washed and fixed in 4% paraformaldehyde solution before flow cytometry
analysis? Please be more precise in the description protocol.

5. M&M Animal protocol: The mice used are the same as the NSG mouse (NOD SCID gamma mouse), a brand of immunodeficient laboratory mice developed and marketed by Jackson Laboratory, which carries the strain 256 NOD.Cg-Prkdcscid Il2rgtm1Wjl/SzJ. NSG branded mice lack mature T cells, B cells, and 257 natural killer (NK) cells. NSG-branded mice are deficient in multiple cytokine signaling pathways and have many innate immunity defects. Please describe the housing condition for these mice.

6. Figure 1. Various human cancer cell lines expressed ROR1 antigen. Flow cytometry was used to confirm the
expression level of the ROR1. Each graphs should show the value of expression of overlay ROR1, secondary only control, and autofluorescence control to provide more meaning analyses of the data. The X-axis should show the log numbers of PE and the Y-axis the scale counts.

7. All flow data should be properly labelled.

8. Figure 2C: How was the the % expression calculated? ROR1 CAR expression rates from 3 independent experiments but showing three scatter data points. Why not show all data points from all experiments?

9. Fig 3D: How was the % cytotoxicity calculated?

10. Fig4D and E: Why not show scatter plot of individual tumor weights?  In Fig 4B and D, why inject CAR T once at day 7? Explain why there is an inflection rise in tumor growth at day 28 for breast tumor and day 16 for the other? \Are there any metastatic disease in these animals following treatment?

11. LDH assay: Please describe the cells used and at what cell numbers?

12. The manuscript shows over 350 grammar errors which require attention.

Round 2

Reviewer 2 Report

Authors responded to the reviewer's comments adequately
